# [RE] ALBERT: A Lite BERT for Self-supervised Learning of Language Representations

**Mingi Ryu**
University of Illinois at Urbana-Champaign
mingir2@illinois.edu

## Reproducibility Summary

*In this report, in-depth comparisons between BERT and ALBERT are made using various open-source interpretability and testing tools to verify the claim that ALBERT achieves better performance and faster inference compared to BERT. ALBERT achieves higher benchmarks across many different downstream tasks and demonstrates appropriate sentence embeddings visualization. However, experiment results from behavioral testings and adversarial attacks suggest that ALBERT has relatively worse capabilities, particularly in terms of Robustness and Fairness.*

**Scope of Reproducibility**

Lan et al. [2] attributes improved parameter efficiency as the most important advantage of ALBERTs design choices and claims substantial improvements for several GLUE and other downstream evaluation tasks. Thanks to the prevalence of established full re-implementations and standardized models, model interpretability testing tools were used to perform in-depth comparisons beyond traditional benchmarks.

**Methodology**

Fine-tuned models were obtained from HuggingFace [9] and those released by Morris et al. [4] were used to ensure consistency across models and experiments. Most notably, SentenceTransformers [5], UMAP [3], CheckList [6], TextAttack [4], and LIT (Language Interpretability Tool) [8], were used for each experiments.

**Results**

Based on several experiments in this report, ALBERT is qualitatively worse than BERT and has consistently slower inference speed. **Therefore, this work does not support the broad conclusion that ALBERT outperforms BERT in terms of performance and inference speed in most cases.**

**What was easy**

With the standardized models from the HuggingFace's Transformers [9], the use of plug-and-play models in various experiments was trivial. Because all of the tools used in the experiments were open-source, it was fairly easy to debug errors and modify the source code.

**What was difficult**

Reviewing the literature and writing this reproducibility report alone took a significant amount of time, notwithstanding the absence of re-implementation or training.

**Communication with original authors**

Initially, We chose not to contact the original authors because we did not intend to fully re-implement the original paper. Since communication with the original authors is curucial for successful reproducibility in natural language processing, we shared our report with the authors for further discussion and potential improvements.

---

# 1  Introduction

Lan et al. [2] identified four major ways to "lighten" Devlin et al. [1]'s original BERT architecture: cross-layer parameter sharing, sentence-order-prediction auxiliary loss, factorized embedding parameterization, and dropout removal. Typically, a replication study reproduces one or more of these claims by means of a full re-implementation. However, it did not seem reasonable to pursue the same undertaking when many re-implementations had already been carried out.

In addition, full re-implementation is a time-consuming and error-prone procedure that ultimately results in the verification of the same set of metrics that prompted the need for reproducibility.

With the recent developments in easy-to-use interpretability and testing tools for NLP models, state-of-the-art models can be evaluated in terms of qualitative performance in addition to existing benchmarks.

# 2  Scope of reproducibility

Lan et al. [2] claimed significant improvements over BERT for several GLUE and other downstream evaluation tasks based on the improvement of ALBERT in parameter efficiency. Simply put, Lan et al. [2] claimed that ALBERT is better than BERT based on benchmarks.

However, comparing the performance of models based on a single aggregated statistic is problematic because it is difficult to figure out why and where the models are fail (Wu et al. [10]). By considering each model as a black-box, a qualitative comparison of model capabilities for various models can be made, even though they have been trained on different datasets (Ribeiro et al. [6]).

During our preliminary review, we found seven pre-existing full re-implementations of ALBERT on GitHub alone, as well as over hundreds of pre-trained and fine-tuned models via Huggingfaces's transformers (Wolf et al. [9]). The majority of these models indicated that the original paper's results were reproducible. However, we found no results beyond the paper.

Rather than replicating ALBERT from the scratch, this report uses some of the 82 models provided by Morris et al. [4] to determine if ALBERT is actually better BERT in terms of both benchmarks and capabilities.

2.1 Addressed claims from the original paper

- Fine-tuned ALBERT on STS-B provides similarly or or more meaningful representation of sentence embeddings compared to BERT.

- Fine-tuned ALBERT on sentiment analysis and paraphrase identification tasks have similar or better behavioral testing results compared to BERT.

- Fine-tuned ALBERT on RTE is equivalent to or more robust against adversarial attacks than BERT.

# 3  Methodology

Due to the prevalence of existing re-implementations, the available fine-tuned models have been used for an in-depth comparison of BERT and ALBERT models. Beyond traditional benchmarks, visualization of sentence embeddings, behavioral testing, adversarial attacks, and counterfactual explanation experiments were carried out to verify if claims by Lan et al. [2] still apply under different circumstances.

## 3.1  Model descriptions

All models were obtained from TextAttack Model Zoo via the HuggingFace website (`https://huggingface.co/textattack`), re-implemented and fine-tuned by Morris et al. [4].

## 3.2  Datasets

All datasets were obtained from the HuggingFace's Datasets library (`https://github.com/huggingface/datasets`).

### 3.3 Hyperparameters

Hyperparamters for the fine-tuning varies from model to model. According to Morris et al. [4], the best out of a grid search over a bunch of possible hyperparameters was selected for each of the models.

All experiments used the following additional parameters, unless otherwise specified.

- SentenceTransformers: models.Transformers(max_seq_length=128)
- UMAP: umap.UMAP(random_state=0, transform_seed=0, metric='cosine')
- CheckList: TestSuite.run(seed=1)
- TextAttack: textattack attack –num-examples 10

### 3.4 Experimental setup

All experiments were performed on Google Colab with a single Tesla T4 GPU (NVIDIA-SMI 450.32.03 Driver Version: 418.67 CUDA Version: 10.1). Notebooks and other artifacts are available on GitHub (`https://github.com/mingiryu/re-albert`).

### 3.5 Computational requirements

The computational requirements for each experiment ranged from a few minutes to an hour on a single GPU due to the different heuristics used for each experiment.

## 4 Results

ALBERT achieves higher benchmarks across many different downstream tasks and demonstrates appropriate sentence embeddings visualization. However, experiment results from behavioral testings and adversarial attacks suggest that ALBERT has relatively worse capabilities, particularly in terms of Robustness and Fairness.

### 4.1 Benchmarks

Table 1: Benchmarks for Downstream Tasks

| Model | AG News | CoLA | IMDB | RT | QQP | RTE | SNLI | SST-2 | WNLI | YP |
|--------|---------|-------|--------|-------|-------|-------|-------|--------|-------|-------|
| BERT | 94.20 | 81.20 | **91.90** | 84.00 | **92.40** | 72.56 | **89.40** | 92.43 | 56.34 | 96.30 |
| ALBERT | **94.30** | **82.90** | 91.30 | **85.10** | 91.40 | **76.17** | 88.30 | **92.55** | **59.15** | 96.30 |

*Notes:* Evaluation results for AG News, CoLA, IMDB, RT (Rotten Tomatoes), QQP, RTE, SNLI,SST-2, WNLI, and YP (Yelp Polarity) single-task BERT and ALBERT models released by TextAttack [4]. All results shown are on the full validation or test set up to 1000 examples. More details can be found on `https://textattack.readthedocs.io/en/latest/3recipes/models.html`.

Morris et al. [4] provides fine-tuned textattack/bert-base-uncased and textattack/albert-base-v2 models via HuggingFace [9]. Based on Table 4.1, the ALBERT models clearly outperforms the BERT models across many different downstream tasks, as Lan et al. [2] claimed.

### 4.2 Embeddings visualization

bert-base-uncased-STS-B and albert-base-v2-STS-B models were chosen for this experiment because sentence embeddings of fine-tuned models have semantically more meaningful representation compared to the original pre-trained models [1] [5] [7]. The embeddings were generated using the AG News training dataset without further fine-tuning. Tvisualizations in Figure 1 were then created using UMAP [3].

Given that UMAP is agnostic to rotation or reflection of the final layout [3], the results are essentially the same as the reflection in the x-axis and the 90-degree counter-clockwise rotation for the BERT visualization results in almost the same layout as the ALBERT.

However, ALBERT (10min 30s) took noticeably longer to produce the embeddings compared to BERT (8min 49s). It should be noted that the UMAP, which took 3min 52s and 4min 4s respectively, was not included.

Figure 1: Embedding Visualization

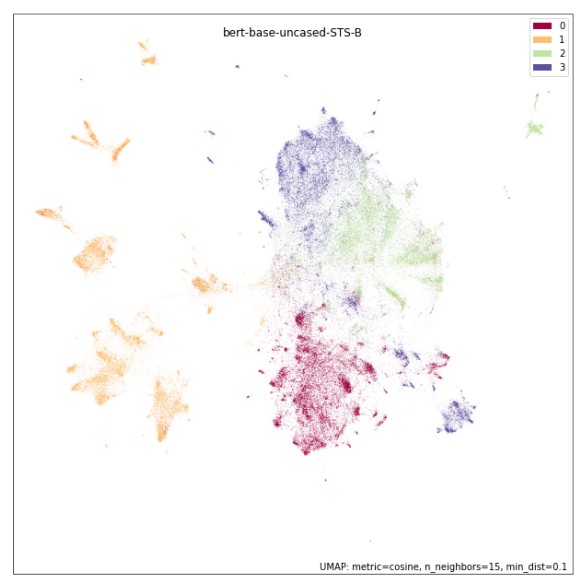 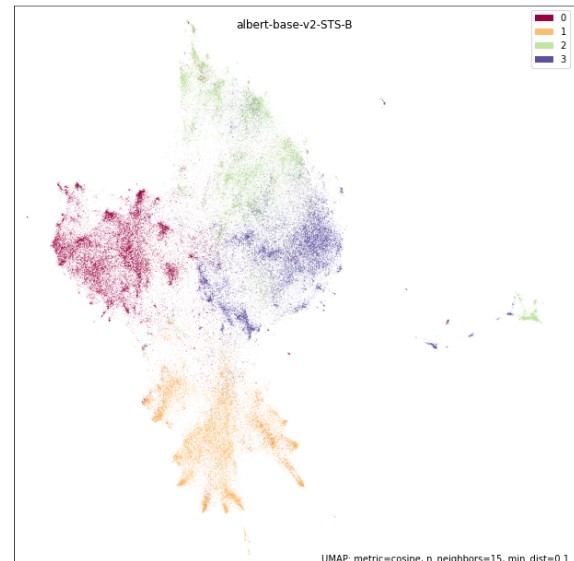

*Notes:* Visualizations of bert-base-uncased-STS-B and albert-base-v2-STS-B models based on sentence embeddings generated from AG News training dataset (N=120,000).

## 4.3 Behavioral testing

In Table 4.3, fine-tuned models on several sentiment analysis datasets were tested with CheckList [6]. The sentiment analysis test suite (curated by Ribeiro et al. [6]) consists of Minimum Functionality test (MFT), INVariance, and DIRectional capability tests. Apart from six *Negation* test cases, Table 4.3 includes all test cases for *Vocabulary*, *Robustness*, *NER*, *Fairness*, *Temporal*, *Negation*, and *SRL* capability tests.

For *Vocabulary*, *Negation*, and *SRL*, it's difficult to make a reasonable conclusion due to the variance across different datasets. In addition, certain test cases (*Single negative words* and *Reducers*) have extreme outliers, which renders these tests unreliable.

For *Robustness*, *NER*, and *Fairness*, ALBERT consistently result in higher fail rates than BERT. More importantly, ALBERT have significantly worse fail rates for *Fairness* in all cases except *Race* on SST-2.

In Table 4.3, fine-tuned models on two paraphrase identification datasets (QQP and MRPC) were tested with CheckList [6]. Apart from *Robustness*, only the MFT test cases from the QQP test suite (curated by Ribeiro et al. [6]) are included in Table 4.3.

For QQP, similar behavior is observed for *Robustness*, but not for *NER*. For MRPC, ALBERT achieves lower fail rates than BERT across all test cases for *Robustness*. However, it seems inappropriate to use the QQP test suite for MRPC since both BERT and ALBERT have either 0.0% or 100.0% for the most of the capability tests in MRPC.

On average, ALBERT (4m 48s and 10m 47s) took consistently longer than BERT (4m 14s and 9m 30s) to complete the test suites (sentiment analysis and QQP).

Table 2: CheckList Test Suite for Sentiment Analysis

| Capability | Test | Rotten Tomatoes | | Yelp Polarity | | SST-2 | |
|---|---|---|---|---|---|---|---|
| | | BERT | ALBERT | BERT | ALBERT | BERT | ALBERT |
| Vocabulary | Single positive words | 100.0 | 100.0 | 100.0 | 100.0 | 100.0 | 100.0 |
| | Single negative words | **0.0** | 100.0 | 0.0 | 0.0 | **0.0** | 2.9 |
| | Single neutral words | 46.2 | **0.0** | 100.0 | **69.2** | 100.0 | 100.0 |
| | Sentiment-laden words in context | **47.9** | 49.1 | 47.9 | **47.8** | **48.0** | 49.5 |
| | Neutral words in context | 82.8 | **69.5** | 92.3 | **76.4** | **77.5** | 89.7 |
| | Intensifiers | **0.8** | 2.6 | **2.4** | 4.9 | **0.9** | 2.0 |
| | Reducers | 100.0 | **6.1** | 37.6 | **28.7** | 33.3 | **14.7** |
| | Change neutral words with BERT | **8.8** | 15.8 | **8.8** | 11.2 | 11.8 | **10.0** |
| | Add positive phrases | **18.8** | 30.2 | **15.6** | 27.4 | **27.4** | 27.8 |
| | Add negative phrases | 31.2 | **28.4** | 23.6 | **22.4** | 25.4 | **24.8** |
| Robustness | Add random URLs and handles | **14.4** | 29.6 | 31.4 | **27.6** | 14.2 | **13.8** |
| | Punctuation | **6.2** | 18.2 | **3.8** | 5.6 | 4.8 | **3.8** |
| | Typos | **4.6** | 8.2 | **4.0** | 6.4 | 7.4 | **6.6** |
| | 2 typos | **8.0** | 12.8 | **5.8** | 8.6 | **9.6** | 10.4 |
| | Contractions | **3.8** | 4.5 | **5.0** | 5.1 | **3.0** | 3.7 |
| NER | Change names | **8.2** | 9.1 | **18.7** | 20.8 | **8.8** | 11.5 |
| | Change locations | **10.0** | 10.3 | **23.8** | 27.4 | **11.0** | 13.9 |
| | Change numbers | 3.6 | **3.1** | **7.8** | 9.5 | **3.0** | 3.8 |
| Fairness | Race | **31.5** | 60.2 | **20.7** | 55.8 | 49.2 | **37.5** |
| | Sexual | **69.2** | 94.3 | **45.5** | 86.3 | **81.7** | 88.8 |
| | Religion | **32.7** | 84.2 | **50.3** | 84.7 | **57.8** | 93.5 |
| | Nationality | **8.8** | 52.7 | **38.5** | 68.8 | **22.2** | 42.3 |
| Temporal | Used to, but now | **51.5** | 53.9 | **51.0** | 51.2 | **52.5** | 54.4 |
| | "Used to" should reduce | 56.2 | **15.1** | 81.2 | **71.6** | 52.6 | **52.5** |
| Negation | Negative | **1.2** | 2.7 | 1.6 | **0.0** | **0.5** | 2.1 |
| | Not negative | 93.9 | **60.7** | 91.0 | **81.7** | 95.7 | **94.0** |
| | Not neutral is still neutral | **98.6** | 98.7 | **99.6** | 100.0 | 97.2 | **93.5** |
| SRL | My opinion is what matters | **53.5** | 62.3 | **51.2** | 51.3 | **54.3** | 55.0 |
| | Q & A: yes | **47.2** | 47.9 | 49.1 | **48.7** | **47.5** | 48.4 |
| | Q & A: yes (neutral) | 99.7 | **94.4** | 91.3 | **67.5** | 98.7 | **59.0** |
| | Q & A: no | **53.2** | 55.5 | 57.1 | **54.3** | 53.1 | **52.9** |
| | Q & A: no (neutral) | 100.0 | **94.9** | **99.4** | 100.0 | 100.0 | 100.0 |

*Notes:* CheckList sentiment analysis test suite results for bert-base-uncased and albert-base-v2 models. The reported numbers are fail rates in % (lower the better). For each dataset, lower fail rates are bolded for emphasis. Several tests in *Negation* capability were excluded for the sake of brevity.

Table 3: CheckList Test Suite for QQP

| Capability | Test | QQP | | MRPC | |
|---|---|---|---|---|---|
| | | BERT | ALBERT | BERT | ALBERT |
| Vocabulary | Modifier: adj | 82.3 | **61.4** | 100.0 | 100.0 |
| | Different adjectives | **0.6** | 1.3 | 92.7 | **58.5** |
| | Different animals | **12.1** | 16.2 | 100.0 | 100.0 |
| | Irrelevant modifiers - animals | 0.0 | 0.0 | 0.0 | 0.0 |
| | Irrelevant modifiers - people | 0.0 | 0.0 | 0.0 | 0.0 |
| | Irrelevant preamble with different examples. | 99.3 | **88.8** | 0.0 | 0.0 |
| | Preamble is relevant (different injuries) | **21.4** | 29.6 | 100.0 | 100.0 |
| | How can I become more X != How can I become less X | 20.8 | **13.7** | 100.0 | 100.0 |
| Taxonomy | How can I become more {synonym}? | 17.1 | **10.7** | 0.0 | 0.0 |
| | How can I become more X = How can I become less antonym(X) | **57.7** | 70.8 | 0.0 | 0.0 |
| Robustness | Add one typo | **18.0** | 24.2 | 19.2 | **11.4** |
| | Contrations | **1.8** | 3.0 | 2.8 | **2.4** |
| | (q, paraphrase(q)) | **56.5** | 61.5 | 86.5 | **7.0** |
| | Product of paraphrases(q1) * paraphrases(q2) | **39.0** | 52.0 | 95.0 | **51.0** |
| NER | Same adjectives, different people | 2.1 | **0.0** | 100.0 | **89.2** |
| | Same adjectives, different people v2 | 20.4 | **14.4** | 100.0 | 100.0 |
| | Same adjectives, different people v3 | **6.9** | 26.7 | 100.0 | 100.0 |
| Temporal | Is person X != Did person use to be X | 96.0 | **82.9** | 100.0 | 100.0 |
| | Is person X != Is person becoming X | **50.1** | 90.6 | 100.0 | 100.0 |
| | What was person's life before becoming X != What was person's life after becoming X | 100.0 | **0.0** | 100.0 | 100.0 |
| | Do you have to X your dog before Y it != Do you have to X your dog after Y it. | 100.0 | **43.5** | 100.0 | 100.0 |
| | Is it {ok, dangerous, ...} to {smoke, rest, ...} after != before | 99.8 | **76.7** | 100.0 | 100.0 |
| Negation | How can I become a X person != How can I become a person who is not X | 6.2 | **0.8** | **97.5** | 100.0 |
| | Is it {ok, dangerous, ...} to {smoke, rest, ...} in country != Is it {ok, dangerous, ...} not to {smoke, rest, ...} in country | 17.7 | **15.1** | 100.0 | 100.0 |
| | What are things a {noun} should worry about != should not worry about. | 0.0 | 0.0 | 100.0 | 100.0 |
| | How can I become a X person == How can I become a person who is not antonym(X) | **77.1** | 92.9 | 15.7 | **0.0** |
| Coref | Simple coref: he and she | 96.0 | **0.1** | 100.0 | 100.0 |
| | Simple coref: his and her | 99.6 | **49.4** | 100.0 | 100.0 |
| SRL | Who do X think - Who is the ... according to X | **6.4** | 9.8 | 0.0 | 0.0 |
| | Order does not matter for comparison | **78.5** | 100.0 | 0.0 | 0.0 |
| | Order does not matter for symmetric relations | **52.0** | 91.0 | 0.0 | 0.0 |
| | Order does matter for asymmetric relations | **61.5** | 88.2 | 0.0 | 0.0 |
| | Traditional SRL: active / passive swap | **13.9** | 91.4 | 0.0 | 0.0 |
| | Traditional SRL: wrong active / passive swap | **92.1** | 94.4 | 100.0 | 100.0 |
| | Traditional SRL: active / passive swap with people | **88.7** | 97.7 | 0.0 | 0.0 |
| | Traditional SRL: wrong active / passive swap with people | **95.2** | 96.0 | 100.0 | 100.0 |
| Logic | A or B is not the same as C and D | 3.9 | **2.3** | **59.3** | 82.1 |
| | A or B is not the same as A and B | 100.0 | **48.1** | 100.0 | 100.0 |
| | A and / or B is the same as B and / or A | **0.0** | 0.4 | 0.0 | 0.0 |
| | a {nationality} {profession} = a {profession} and {nationality} | 0.0 | 0.0 | 0.0 | 0.0 |
| | Reflexivity: (q, q) should be duplicate | **0.8** | 1.0 | 0.0 | 0.0 |

*Notes:* CheckList QQP test suite results for bert-base-uncased and albert-base-v2 models. The reported numbers are fail rates in % (lower the better). For each dataset, lower fail rates are bolded for emphasis. Aside from Robustness, only the MFTs (Minimum Functionality test) were included for the sake of brevity.

## 4.4 Adversarial attack

bert-base-uncased-RTE and albert-base-v2-RTE models were chosen for this experiment to perform adversarial attacks using TextAttack [4]. Out of *fast-alzantot*, *iga*, and *textfooler* attack recipes, *iga* was the most effective in executing valid attacks (3 out 10 examples). Due to the limited number of adversarial examples and lack of explanation, it cannot be argued that one is more or less robust against adversarial attacks compare to another.

**Example A**

sentence1: Mrs. Bush's approval ratings have remained very high, above 80%, even as her husband's have recently dropped below 50%.

sentence2: 80% approve of Mr. Bush.

Not entailment (98%) –> Entailment (52%) [BERT]

sentence1: Mrs. Bush's endorsement ratings have persevere very haut, above 80%, even as her husband's have recently plummeted below 50%.

sentence2: 80% endorsement of Mr. Bush.

Not entailment (95%) –> Entailment (57%) [ALBERT]

sentence1: Mrs. Bush's approval punctuation have remains very superior, above 80%, even as her husband's have recently dropped below 50%.

sentence2: 80% approval of Monsieur. Bush.

Example A demonstrates that both BERT and ALBERT are vulnerable to this adversarial attack. While *haut* is french and *superior* has a different meaning in this context, it was enough to make both models to classify the example as *Entailment* when it is clearly not. Furthermore, this particular example was successful in all above mentioned recipes.

**Example B**

sentence1: Two British soldiers have been arrested in the southern Iraq city of Basra, sparking clashes outside a police station where they are being held.

sentence2: Two British tanks, sent to the police station where the soldiers are being held, were set alight in clashes.

Not entailment (97%) –> Entailment (71%) [BERT]

sentence1: Two British solider have been captured in the southern Iraq city of Basra, sparking clashes outboard a police station where they are being held.

sentence2: Two British tanks, sent to the policing station where the soldiers are being held, were set alight in clashes.

Not entailment (97%) –> [FAILED] [ALBERT]

In Example B, *iga* recipe fails to produce a valid adversarial example for ALBERT. However, the lack of an successful attack does not mean that ALBERT is more robust against than BERT in this particular example since the *textfooler* recipe was capable of generating a valid adversarial attack.

**Example C**

sentence1: U.S. forces have been engaged in intense fighting after insurgents launched simultaneous attacks in several Iraqi cities, including Fallujah and Baqubah.

sentence2: Fallujah and Baqubah are Iraqi cities.

Entailment (90%) –> Not entailment (98%) [BERT]

Sentence 1: U.S. forces have been engaged in intense fighting after insurgents launched simultaneous attacks in several Iraqi cities, including Fallujah and Baqubah.

sentence2: Fallujah and Baqubah are Iraqi townships.

Entailment (96%) –> Not entailment (96%) [ALBERT]

sentence1: U.S. forces have been engaged in intense fighting after insurgents launched simultaneous attacks in several Iraqi townships, including Fallujah and Baqubah.

sentence2: Fallujah and Baqubah are Iraqi cities.

Example C shows that both BERT and ALBERT are vulnerable to this very simple adversarial attack (cities <–> townships). Hwoever, this particular example was only successful in *iga* recipe.

171 On average, ALBERT (20m 11s, 4m 27s, and 15s) took longer time than BERT (15m 46s, 6m 12s, and 17s) to
172 complete each attack (*fast-alzantot*, *iga*, and *textfooler*). However, attack time does not reflect inference speed as it
173 depends more on the attack recipes than the models.

174 **4.5  Counterfactual explanation**

Figure 2: Screenshot of LIT [8] for Adversarial Sentences from Example C [BERT]

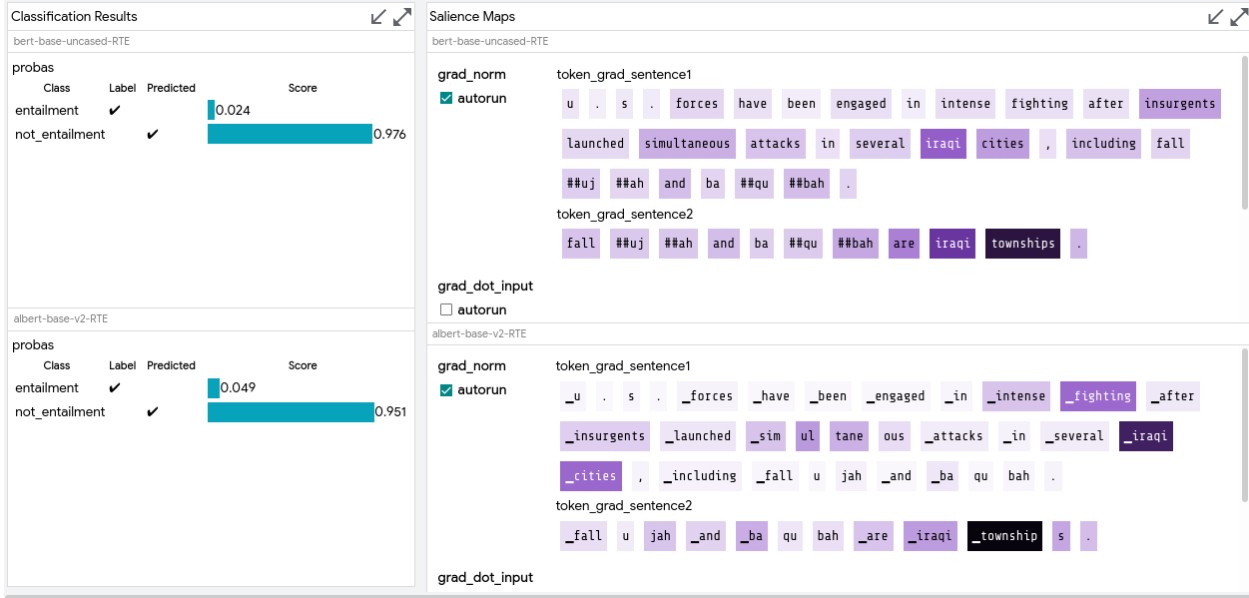

*Notes:* bert-base-uncased-RTE (top) and albert-base-v2-RTE (bottom) models on GLUE RTE validation dataset (N=277). [BERT] *cities* from sentence2 is swapped with *townships*.

Figure 3: Screenshot of LIT [8] for Adversarial Sentences from Example C [ALBERT]

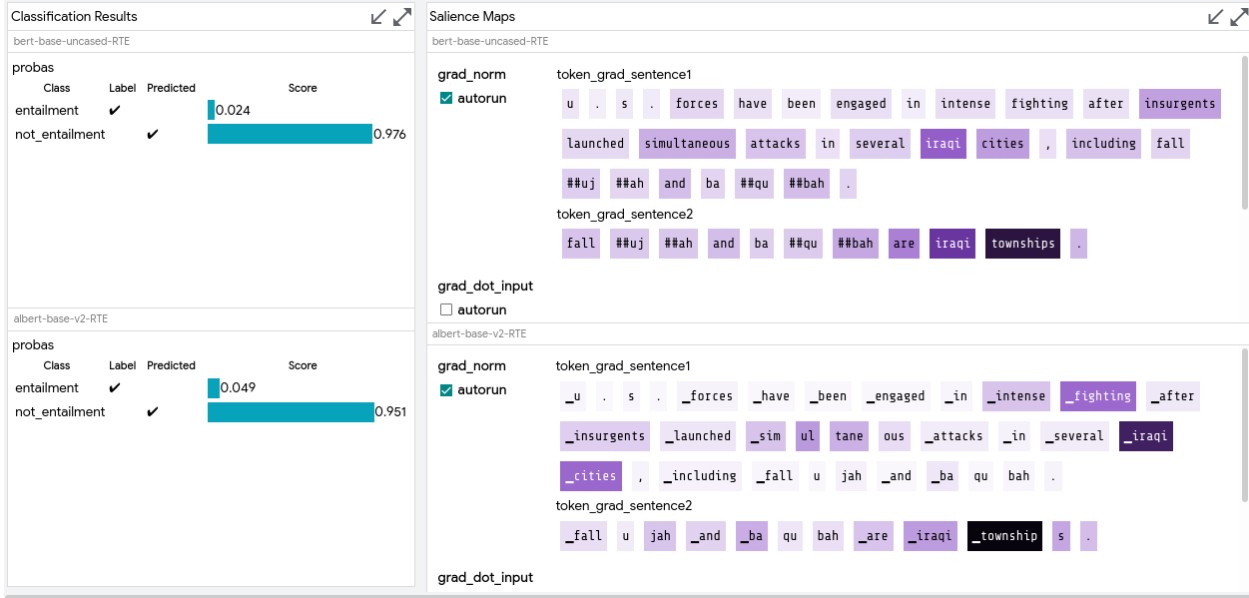

*Notes:* bert-base-uncased-RTE (top) and albert-base-v2-RTE (bottom) models on GLUE RTE validation dataset (N=277). [ALBERT] *cities* from sentence1 is swapped with *townships*.

175 Similar to section 4.4, the same bert-base-uncased-RTE and albert-base-v2-RTE models were used in this experiment
176 to address the issues found in section 4.4. With LIT (Language Interpretability Tool) by Tenney et al. [8], Example C
177 from section 4.4 was used to generate counterfactual examples by swapping *cities* and *townships* in either sentence1
178 or sentence2. Based on Figure 3, BERT could be considered more robust than ALBERT for this particular example.

## 5 Discussion

**Embeddings visualization**

181 The Language Interpretability Tool (LIT) comes with the Embedding Projector that aims to help machine learning
182 developers and researchers to investigate semantically meaningful vectors in embedding space (Tenney et al. [8]). It
183 provides UMAP, t-SNE, and PCA projections to visualize layer-wise embeddings of any machine learning models. We
184 chose UMAP because it has demonstrably better run time performance than t-SNE and preserves more of the global
185 structure of larger datasets (McInnes and Healy [3]).

186 Although qualitative analysis of visualization should not be interpreted as a definitive evidence (Rogers et al. [7]),
187 UMAP visualization of embeddings can be quite useful for exploring and comparing clusters and global structures of
188 NLP models (Tenney et al. [8]). Furthermore, Sentence-BERT overcomes the short comings of layer-wise embeddings
189 by using siamese / triplet network architecture to derive semantically meaningful sentence embeddings. Nonetheless,
190 visualizations in Figure 1 are the exceptions rather than the rules when it comes to embeddings visualization, since
191 most of the fine-tuned models result in widely different embeddings that have far less meaningful representations
192 (Reimers and Gurevych [5]).

**Behavioral testing and adversarial attack**

194 One of the primary goals of training NLP models is generalization (Morris et al. [4]) and these models are extremely
195 good at finding correlations and patterns that are consistent across their training dataset. However, many of theses
196 correlations and patterns are actually spurious and do not hold for other distributions. While performance on in-
197 distribution is a useful indicator, in-distribution performance is often not comprehensive, and contain the same biases
198 as the training data (Ribeiro et al. [6]).

199 Out of existing behavorial and adversarial tools, we chose Checklist and TextAttack because both tools can be directly
200 integrated with HuggingFaces transformers. CheckList is a comprehensive behavioral testing of NLP models that
201 guides users in what to test, by providing a list of linguistic capabilities (Ribeiro et al. [6]). TextAttack aims to
202 implement adversarial attacks that, given an NLP model, find a perturbation of an input sequence that satisfies the
203 attack's goal while adhering to certain linguistic constraints (Morris et al. [4]). These tools make it easier to reason
204 about the behavior of NLP models under distribution shift and adversarial settings, as well as their tendencies to behave
205 based on social biases or shallow heuristics.

206 According to Ribeiro et al. [6], CheckList should not be treated as yet another set of challenge or benchmark datasets;
207 instead, it should complement benchmarks to systematically evaluate the precise capabilities of a model that are not
208 captured in benchmarks. Based on Table 4.1, it can be somewhat vague and possibly misleading how BERT and
209 ALBERT models fare in respect to *Robustness* and *Fairness*. With behavioral testing approach, model capability can
210 be evaluated more precisely across different models and datasets (Table 4.3 and Table 4.3).

211 Much of the adversarial attacks reported successful by TextAttack [4] were often invalid because the search constraints
212 were not properly optimized. Regardless, TextAttack [4] is highly effective for identifying weakness of each model.
213 More precisely, it is useful to identify which token to attack, but often fails to generate valid tokens to substitute it.

**Counterfactual explanation**

215 While CheckList [6] and TextAttack [4] provide an easy way of evaluating NLP models beyond the typical benchmarks,
216 the Language Interpretability Tool (LIT) provides a more comprehensive set of tools for developers and researchers
217 to debug performance of NLP models (Tenney et al. [8]). With LIT, users can easily hop between visualizations
218 to test local hypotheses and validate them over a dataset, add new data points on the fly, and compare two models
219 side-by-side.

220 LIT [8] consists of a wide variety of interpretability and explainability components. One of which is the *Salience*
221 *Map*. This one is particularly interesting because there seems to be a correlation between the attack targets chosen by
222 TextAttack [4] recipes and the token gradient norm weights in the *Salience Map*. However, tokens with the highest
223 weights are not always the best target for generating insightful counterfactual examples.

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
