# OpenReview forum: "[RE] ALBERT: A Lite BERT for Self-supervised Learning of Language Representations"
_ML_Reproducibility_Challenge/2020 — Reject_

### Official Review · AnonReviewer3 · 2021-02-22
**ALBERT: A Lite BERT for Self-supervised Learning of Language Representations**

**Rating:** 8
**Confidence:** 5

**Review:**

This paper can be accepted without any modifications.

**Familiar With The Original Paper:**

I have read the original paper

**Reproducibility Summary:**

Report has summary

---

### Official Review · AnonReviewer1 · 2021-02-28

**Rating:** 2
**Confidence:** 5

**Review:**

# Reproducibility Summary

The summary is provided, although given that the results indicate ALBERT's findings were not reproduced, I think that the "what was difficult" section could have been written with more details. In other words, if the findings of the paper are refuted there should be substantial effort demonstrating good faith reimplementation and/or reasons why the original findings are wrong.

It's also unclear whether the original findings were factually incorrect — they reported an incorrect number — or were overreaching in their claims.

# Scope of Reproducibility

The scope is two-fold:
- Reproduce the ALBERT model from a pre-trained checkpoint.
- Compare ALBERT performance to other baselines using more in-depth analysis.

# Communication

No communication was mentioned. EDIT: The reproduction authors do explain why they chose not to communicate.

# Hyperparameter Search

I think this section could use substantial improvement. Fine-tuning tends to require very careful hyperparameter choice in order to prevent issues such as "catastrophic forgetting". Also, it seems the authors use off-the-shelf fine-tuned models and did not perform their own fine-tuning. This is fine, but much more due diligence should be done to explain how the checkpoints they use were created.

# Ablation Study

The reproduction authors did not do any re-training with different hyperparameters or other ablation. Although they did run a new set of evaluation.

# Discussion

I think this paper would benefit from richer discussion. Specifically, it would be interesting to explain in detail why the evaluation methods used were chosen.

# Recommendations

The reproduction authors did not provide a recommendation to the original authors.

# Results beyond the paper

The reproduction authors were ambitious and set out to report many results and findings not in the original work. The line of inquiry is interesting — perhaps leaderboards like GLUE do not reflect all important model properties, such as fairness and robustness.

# Overall Organization and Clarity

Several parts of the paper were not clear. Even if this is a reproducibility paper, the authors should still summarize in detail the paper they are reproducing and any other techniques (such as TextAttack). This would help, but other parts of the writing were a bit ambiguous. For instance:

> However, ALBERT (10min 30s) took noticeably longer to produce the embeddings compared to BERT (8min 49s).

Does this mean that UMAP took 10m30s, or that ALBERT took 10m30s to output the embeddings used for visualization?

In addition, it would be helpful for the reader if the plots and figure were embedded more naturally in the text. It can be jarring when a crucial table takes a full page, and perhaps it could be broken into smaller sections and distributed throughout the text adjacent to its mentions.


**Familiar With The Original Paper:**

I have read the original paper

**Reproducibility Summary:**

Report has summary

---

### Official Review · AnonReviewer2 · 2021-03-04
**[RE] ALBERT: A Lite BERT for Self-supervised Learning of Language Representations**

**Rating:** 3
**Confidence:** 4

**Review:**

This paper reported on a comparative study of BERT and ALBERT for self-supervised learning in the context of language representations, concluding that the authors were unable to reproduce empirical evidence for the claims presented by the selected paper. However, in their view "communication with the original authors did not seem appropriate, as this study does not seek to fully re-implement the original paper." I find that this is an unfair approach to the selected paper, as in order to act politically and respectfully among our scholarly community, contacting the first/corresponding author would have been the right first action here. As concluded in https://www.aclweb.org/anthology/W16-6110.pdf , https://www.ncbi.nlm.nih.gov/pmc/articles/PMC5998676/ , and https://www.aclweb.org/anthology/R19-1089.pdf, for, successful reproducibility in natural language processing requires more than having access to data and code, and in general, contacting the authors leads to obtaining the same main findings.

**Familiar With The Original Paper:**

I have not read the original paper

**Reproducibility Summary:**

Report has summary

---

### Decision · Program_Chairs · 2021-03-31

**Decision:**

Reject

**Comment:**

The results of the report may diverge from the original paper because out-of-the-box HuggingFace models may not be exactly the same as the original paper in terms of implementation. Furthermore, the authors did not do their due diligence (including hyperparam tuning) for a fair comparison.